# Properties of “Stable” Mosquito Cytochrome P450 Enzymes

**DOI:** 10.3390/insects16020184

**Published:** 2025-02-08

**Authors:** George Tzotzos

**Affiliations:** Visiting Research Fellow, Department of Agricultural, Food and Environmental Sciences, Marche Polytechnic University, 60100 Ancona, Italy; gtzotzos@me.com

**Keywords:** *A. gambiae*, *A. aegypti*, *C. quinquefasciatus*, CYP, cytochrome P450 enzymes, genomic properties, protein properties

## Abstract

Cytochrome P450 enzymes (CYPs) are a superfamily of enzymes found in all kingdoms of life. Insects, in particular mosquitoes, constitute the taxon with the most P450 genes. While some CYPs have arisen from dynamic evolutionary events (“labile” CYPs), others comprise only one or two members, which have changed little in the course of evolution (“stable” CYPs). Although CYPs are known to be involved in essential biosynthetic and developmental processes, the precise function of most CYPs has not been established experimentally. The main hypothesis of the current study is that CYPs carry an evolutionary imprint reflected in their sequences, and by extension their function. Here, a bioinformatic analysis of CYPs showed that “stable” and “labile” mosquito P450s can be differentiated on the basis of a number of genomic and sequence features. “Stable” CYPs are encoded by longer genes with more exons than “labile” CYPs, and the corresponding proteins are enriched in hydrophobic amino acids. Functional prediction linked a large number of “stable” CYPs to biosynthetic and developmental functions.

## 1. Introduction

Mosquitos are infectious disease vectors having a profound impact on human health [1]. *Anopheles* mosquitoes transmit the parasite *Plasmodium falciparum*, the causative agent of malaria; *Aedes aegypti* is the vector of the transmission of yellow fever; and *Culex* mosquitoes transmit West Nile virus and Saint Louis encephalitis virus, as well as the nematode that causes lymphatic filariasis. In the absence of vaccines for these diseases, insecticides are widely applied as a means of mosquito control. This strategy has limitations in that insecticides are not only highly ecotoxic but are also becoming increasingly ineffective due to acquired resistance of the targeted insects. Resistance is mediated not only by mutations in the target insect receptors but also through insecticide detoxification [2]. Cytochrome P450 enzymes have been shown to be essential for insect development and survival, but probably a larger number of P450s are involved in insecticide metabolism and resistance [3].

Cytochrome P450 enzymes, also known as CYPs, are haem-containing monooxygenases found in all kingdoms of life [4]. They first appeared in the scientific literature in 1958 [5]. The Pfam database of protein families and domains classifies CYPs as a distinct family of enzymes (Pfam entry PF00067) involved in the oxidative degradation of various compounds [6]. In P450s, the haem iron forms a pentacoordinate system with the axial sulphur of a conserved cysteine (Cys) protein amino acid residue in the active site of the enzyme [7]. The conserved Cys residue exists as a thiolate anion giving a characteristic absorption band of 450 nm, which is also typical of the spectral properties of other cytochromes such as the b-type haem-protein haemoglobin and myoglobin. However, unlike these cytochromes, CYP monooxygenases are not electron-transfer proteins and function in the presence of redox partners, usually NAD(P)H-dependent ferredoxin or FAD- and FMN- containing CPR-type reductases by transferring molecular oxygen to -CH, -NH, and -SH moieties of substrates with the concomitant reduction of the oxygen atom to water [8]. Consequently, a more appropriate term for CYP enzymes would be P450 ‘haem-thiolate’ or P450 monooxygenase proteins rather than cytochrome P450s.

CYPs show an extraordinary diversity in their reaction chemistry. In mammals, they participate in oxidative, peroxidative, and reductive metabolism of numerous endogenous compounds such as fatty acids, cholesterol, steroids, retinoids, vitamin derivatives, bile acids, porphyrins, thromboxane A2, prostacyclins, eicosanoids, and other lipid mediators. In plants, they are involved in the secondary metabolism of, amongst others, phenolic compounds, alkaloids, and gibberellins. The diversity of biosynthetic reactions catalysed by P450 is reviewed by Fujiyama, et al. [9]. Last but not least, CYPs metabolise an enormous range of xenobiotics and endobiotics including drugs, insecticides, environmental chemicals, and pollutants, as well as natural plant products, and bacterial metabolites. P450s are involved in the metabolic detoxification of xenobiotics (phase I metabolism). This involves the addition or unmasking of polar groups, such as hydroxy, amine, or sulphydryl groups, in the xenobiotic substrate followed by hydrolysis, oxidation, or reduction. The resultant reaction intermediates are then further metabolised by phase II enzymes, mainly glutathione-S-transferases [10], and transported into the extracellular space through interactions with transmembrane proteins (phase III), mainly ATP-Binding Cassette (ABC) Transporters [11].

By 2018, more than 300,000 CYP sequences had been mined and collected in all areas of the tree of life [12] but the precise function of individual CYP proteins remains largely unknown. It is estimated that only 0.2% of the genes deposited in different databanks have been functionally characterised [13,14]. CYPs were first classified into distinct groups by Nerbert and Gonzalez in 1987. Following phylogenetic criteria, gene organisation, and sequence similarity, CYPs are grouped into kingdom-specific clusters, named clans, each of which represent genes that diverged from a single common ancestor and can include one or multiple families [15,16]. Generally, groups of proteins having amino acid identity over 40% are assigned to the same family, whereas proteins having identity above 55% are allocated to the same subfamily. Gene families that repeatedly cluster in the same phylogenetic clade are grouped into the same clan. In this classification scheme, a family-specific number is given after the root symbol CYP, followed by a letter and a number indicative of the subfamily and the gene, respectively. Clans can consist of one or multiple families and are given the name of the smallest family number present in the clade. For example, if a single clade in a phylogenetic analysis comprises the CYP7, CYP8, and CYP39 families, these families become part of clan 7 [17]. Clan-based classification can be volatile and the addition of new, more distant sequences can lead to interleaved branches obscuring the boundaries between different classes [18].

Amongst animals, insects constitute the taxon with the most P450 genes, which are grouped into six clans, namely, CYP2, CYP3, CYP4, CYP16, CYP20, and mitochondrial (mito). Two clans, CYP16 and CYP20, are restricted to certain species in Apterygota and Paleoptera [14]. Mosquitoes, second only to the deer tick (*Ixodes scapularis*), have a disproportionately large number of P450 genes [19,20]. CYP protein-coding genes in the diptera *D. melanogaster* (*Dmel*), *A. gambiae* (*Agam*), *A. aegypti* (*Aaeg*), and *C. quinquefasciatus* (*Cqui*) form groups in clans 2, 4, and 6 and the mitochondrial clan. The number of CYPs populating each clan vary considerably (Table 1).

Gene assignment to clans and families may change with each new genome release and be further complicated by the fact that genomes can consist of different numbers of CYPs found in natural populations [23]. CYP assignment discrepancies may also arise from genome annotations differing in the coverage and quality of assembly of sequenced genomes, or due to homozygosity and polymorphisms in gene copy numbers [20,24].

Genes of the CYP2 and mitochondrial clans are relatively conserved and they form many families with few or even single members. Members of these clans have been shown to participate in core developmental and physiological functions and their evolution is well conserved with many families having few or even single members [18,25]. CYP clans 3 and 4 are by far the most populated due to the lineage-specific gene amplification of paralogs or “phylogenetic blooms” [23,26]. These “blooms” are thought to result in response to environmental stimuli and, in particular, to the selection pressure exerted by polluted habitats and, in particular, due to the widespread application of chemical insecticides [21,27].

Phylogenetic studies have shown that the expansion of the P450 gene repertoire is shaped by gene duplication, gene birth and death, and gene neo-functionalisation [23]. These events are driven by selection acting on what is available at the time and can be described by a power law typical of simple birth and death models. An implication of this model is that CYPs may switch their function from physiology to detoxification and vice versa [19] or that some CYP genes may be functionally redundant [28].

Yet, some CYPs may deviate from this stochastic evolutionary model and a small number of CYPs have been shown to be “stable” in evolutionary terms, as, for example, in *D. melanogaster*, where 31 such “stable” genes have been identified [29]. Amongst these are the so-called “Halloween” genes, such as *spook* (*spo*), *phantom* (*phm*), *disembodied* (*dib*), *shadow* (*sad*), *spookier* (*spok*), and *spookiest*. These genes are essential for the biosynthesis of moulting hormones [30]. Functional investigation in different tissues of *D. melanogaster*, using RNAi screens, revealed that knockdown of nine of these CYPs resulted in lethality [31]. In light of this evidence, it has been proposed that natural selection may also act on copy number polymorphisms within a species. Although this “selectionist” model has been contested, it is agreed that “conserved orthologous genes” are present in some phylogenetic branches of CYPs [14].

Regardless of the evolutionary model, it is still not possible to distinguish phylogenetic branches related to essential physiological and developmental functions from those related to detoxification. Even in the case of *D. melanogaster*, which serves as a model organism, well over 60% of the total number of P450s are an “orphan” in terms of functional characterisation [32]. Likewise, phylogenetic branches of CYPs cannot be assigned ubiquitous xenobiotic detoxification functions as (i) different CYPs respond to insecticide selection in different mosquito populations and geographical regions [33], and (ii) CYPs are also known to play an indirect role in insecticide resistance [34,35]. As a result, susceptible and resistant strains with a common genetic background are usually not available to compare, and susceptible reference strains are biassed by geographic variation or genetic drift. This makes attempts to identify one or a few metabolic genes associated with insecticide adaptation difficult or even futile [36,37].

“Gene essentiality” refers to those genes where a single gene-knockout results in lethality or severe loss of fitness. Earlier work established that essential and non-essential genes can be recognised on the basis of their respective physical, chemical, and evolutionary properties [38,39,40]. Based on this, bioinformatic methods were used to differentiate “stable” from “labile” mosquito P450s. Given that “stable” CYP genes may also be involved in auxiliary, non-essential metabolic networks, the analogy with “gene essentiality” may be tentative.

## 2. Materials, Methods, and Datasets

CYPs identified in the sources specified in Table 1 were submitted to VectorBase (release 68) [41] to source genomic and protein data, and CYPs with ambiguous annotations were removed. VectorBase hosts over 20 sequenced and annotated mosquito genomes and curates experiment metadata [42]. 

The total number of CYPs in the dataset was 486 (Appendix A). These were mapped onto 83 ortholog groups derived from OrthoMCL Release 6.21 [43,44]. The latter is a genome-scale algorithm for the identification of orthologous protein sequences and provides not only groups shared by two or more species/genomes, but also groups representing species-specific gene expansion families [43,44]. The dataset was further split into two subsets, one of which included CYPs with zero or up to three paralogs and one that included CYPs with four or more paralogs (Appendix A). The division is based on earlier work with *D. melanogaster* CYPs in which genes with up to three duplications were shown to derive from evolutionarily stable clades, and genes with more than three duplications were from “dynamic/unstable” clades. In this paper, the term “labile” is used in preference to “dynamic/unstable” [29].

Redundancy reduction: Each subset was purged by removing genes/proteins having sequence similarity > 80%. Redundancy reduction is necessary to avoid undesirable bias in a statistical analysis arising from the presence of multiple copies of similar genes/proteins [45]. SkipRedundant of the EMBOSS suite of bioinformatics tools [46] was used to obtain non-redundant (culled) datasets, one from the “stable” and “labile” subsets. The numbers of CYPs in each of the resulting subsets are summarised in Table 2.

Gene and protein sequence-based features: The gene, protein, and functional features and bioinformatic methods used to characterise “stable” and “labile” P450 are summarised in Table 3.

Protein sequence properties: The length of our “stable” and “labile” proteins was obtained from VectorBase. The EMBOSS programme Pepstats [46] was used to obtain statistics of a number of properties of FASTA formatted protein sequences. These attributes include molecular weight, number of residues, charge, isoelectric point, and amino acid composition. Hydrophobicity was determined by multiplying the frequency of each amino acid in each P450 with its Kyte and Doolitttle hydropathicity index [50] and summing up the resulting values.

Pepstats groups amino acids into nine categories: Tiny (A, C, G, S, and T); Small (A, B, C, D, G, N, P, S, T, and V); Aliphatic (I, L, and V); Aromatic (F, H, W, and Y); Non-polar (A, C, F, G, I, L, M, P, V, W, and Y); Polar (D, E, H, K, N, Q, R, S, T, and Z); Charged (B, D, E, H, K, R, and Z); Basic (H, K, and R); and Acidic (B, D, E, Z). Pepstats was run with the default parameters’ setting. MATLAB scripts were used to extract feature values from the output files and determine corresponding statistical values (MATLAB. (2024). Version R2024a. Natick, MA, USA: The MathWorks Inc.).

Genomic properties: Genomic data (gene, CDS, intron/exon lengths) were obtained from VectorBase. %CG content data were obtained from the Ensembl Metazoa, release 60. For genes with multiple transcripts, the longest-length transcript was used to determine the number of exons and total exon length. The intron length of a gene was calculated by subtracting the total exon length from the corresponding gene length.

Phosphorylation: Protein phosphorylation was predicted using PhosNet 3.0 [47]. The programme predicts serine, threonine, or tyrosine phosphorylation sites in eukaryotic proteins using neural networks. Kinase-specific predictions were used for serine and threonine, and generic predictions were used for tyrosine. In all cases, the confidence score used was >0.7.

Signal peptides, transmembrane domains, and protein localisation: Signal peptides, transmembrane domains, and protein subcellular localisations were predicted using BUSCA [48]. The programme runs as a web server, which integrates several resources to predict subcellular localisation including protein feature predictors (DeepSig, TPPred3, PredGPI, BetAware, and ENSEMBLE3.0) and protein localisation predictors (MemLoci, BaCelLo, SChloro). In addition, BUSCA annotates relevant protein features, such as signal/transit peptides, GPI anchors, and transmembrane domains.

Conserved P450 motifs: Conserved P450 signature motifs were detected using STREME, part of the MEME Suite of bioinformatics tools [51]. STREME discovers ungapped motifs that are enriched in user-submitted sequences. The default settings were used.

Gene Ontology terms: GO term enrichment was obtained using (a) the STRING database [52], g:Profiler [53], DAVID [54], and NetGO 3.0 [49]. The STRING database integrates information from genomic, functional, and structural databases to enhance the prediction and interpretation of interactions. It also provides functional annotations, such as GO terms (Gene Ontology), KEGG pathways, and other biological categories to help interpret the biological relevance of interactions. g:Profiler is a suite of programmes, one of which, g:GOSt, maps the submitted gene lists to different sources of data in order to extract annotations of significantly enriched metabolic pathways, and processes. DAVID is a popular bioinformatics resource system for the functional annotation and enrichment analysis of genes. There are several applications within DAVID allowing functional annotation clustering and gene functional classification. NetGO 3.0 predicts molecular function ontology from sequences using protein language models. NetGO addresses shortcomings of GO functional annotations and is considered as one of the best methods at predicting function ontology [55,56]. The contextual visualisation of the GO terms was performed using GOATOOLS [57].

Statistical analysis: Statistical tests were carried out throughout using the statistics toolbox of MATLAB. The sequence properties did not show a normal distribution and the statistical significance of each property was determined using the two-tailed nonparametric Wilcoxon rank sum test. The test is equivalent to a Mann–Whitney U test. Statistical significance was determined at the 0.05 level and the Bonferroni correction was applied to calculate corrected *p*-values. The Chi-squared (χ^2^) test was also carried out to check whether the frequencies of a particular feature in “stable” and “labile” genes differ from each other.

## 3. Results

### 3.1. Analysis of Genomic Features

Gene length, %CG content, transcript and exon numbers, as well as intron and exon lengths were analysed in all datasets. The results are shown in Table 4. Compared to “labile” P450s, “stable” genes were shown to be generally longer and have a greater number of exons. The opposite is true for intron lengths, where introns in the “labile” dataset are on average considerably shorter (Table 4). The same trends are observed in the subsets of “culled” genes (Figure 1). The GC content of the “stable” CYPs is less than that of the “labile” CYPs but the difference may not be statistically significant.

### 3.2. Analysis of Protein Features

The protein average molecular weight, charge, isoelectric point, and frequencies of different amino acid categories were analysed statistically to determine significant differences in the two groups of P450 (Table 5). “Labile” proteins were found to have greater proportions of acidic, basic, charged, and polar amino acids. In the case of charged amino acids, the *p*-value in the culled dataset was marginally above the Bonferroni correction. “Stable” proteins had a higher median value for aliphatic amino acids and hydrophobicity. Differences between the protein features examined follow the same trend in the culled datasets (Figure 2).

Variations in the frequencies of amino acids found in “stable” and “labile” P450s are shown in Table 6. Statistically significant variations were observed in 11 amino acids (8 in the culled dataset). The relative proportion of Cys, Arg, Leu, and Trp is higher in “stable” P450s, and that of Glu, Lys, and Met is higher in “labile” proteins. Variations in four of these (Ala, His) were not statistically significant in culled datasets of proteins.

### 3.3. Phosphorylation

Protein phosphorylation plays crucial roles in the regulation of cellular and metabolic processes such as cell differentiation and cell division. Cytochrome P450s have been shown to be subject to phosphorylation mediated by different protein kinases, cAMP-dependent protein kinase A being the most prominent one. In insects, phosphorylation is thought to play a role in the control of genotoxic metabolites [58]. P450 phosphorylation was predicted using NetPhos-3.1 [47] (Appendix A). The median number of phosphorylated residues was three with the exception of threonine in the unculled “labile” dataset (median = 4). The statistical differences between “labile” and “stable” P450s in the culled and unculled datasets were not statistically significant.

### 3.4. Signal Peptides, Transmembrane Domains, and Protein Location

Insect P450s are transmembrane proteins bound either to the endoplasmic reticulum (ER) or the inner mitochondrial membrane by means of the N-terminal transmembrane helix (TMH) [59]. In mitochondrial CYPs, the N-terminal anchor is missing. Instead, a topogenic sequence present at the nascent enzyme serves as a signal peptide to ensure that the enzyme is transported and incorporated into the mitochondrial membrane [60]. Both transmembrane and mitochondrial P450s share the same fold and very similar tertiary structures [61]. The transmembrane domains usually adopt an α-helical structure while passing through the lipid bilayer once (single-pass proteins) or multiple times (multiple-pass proteins). In non-mitochondrial P450s, the catalytic domain of the enzyme is on the cytosolic side of ER and the N-terminus *α*-helix protrudes on the luminal side of ER. This posture of CYPs in relation to the membrane is stabilised by hydrophobic interactions between the residues of the N-terminal helix and the lipophilic ER environment. The N-terminal anchor has been associated with CYP trafficking into the ER or mitochondria, in interactions with different phospholipids, and as a mediator of CYP heteromer formation.

Signal peptides, subcellular localisation, and the number of transmembrane helices were predicted using BUSCA [48] (Appendix A). A summary of the results is given in Table 7. All CYPs were predicted to have at least two transmembrane domains. Approximately 20% of CYPs, both in the unculled and culled datasets, were predicted to lack the N-terminal helix. The differences between the datasets were not statistically significant.

### 3.5. Conserved Motifs

The identification of P450s in insects is based on a protein sequence analysis, especially the presence of five P450 signature motifs that are relatively well conserved. These are arranged from the N- to the C-terminal and are known as helix C (WxxR), helix I (GxE/DTT/S), helix K (ExLR), PERF (PxxFxPE/DRE), and haem-binding motifs (PFxxGxRxCxG/A) [62,63]. The haem-binding motif includes the cysteine (thiolate) ligand to the haem iron. In the motif pattern definition above, the symbol ‘x’ is used for a position where any amino acid is accepted, and ambiguities are indicated by listing the acceptable amino acids for a given position separated by ‘/’.

The MEME Suite [51] was used for the identification of differences between the culled and non-culled datasets. Sequence logos for the PERF and the haem-binding motifs are summarised in Table 8. Culling has no effect on the composition of amino acids of the two motifs. The amino acid composition of the PERF motif, in both the “stable” and the “labile” datasets, shows little or no difference in the frequencies of amino acids of the variable positions of the motif. On the other hand, the “labile” and “stable” datasets exhibit significant differences in the amino acid composition of the haem motif. In the “labile” datasets, the variable positions of the motif are occupied by Ser (S), Ala (A), Pro (P), Arg (N), Ile (I), and Gly (G) with 100% frequency of occurrence. This is not the case in the ”stable” datasets where the variable positions are occupied by diverse amino acids.

### 3.6. Gene Enrichment Analysis

Gene Ontology (GO) [64] enables the classification of gene functions through the application of controlled vocabularies (ontology) to annotate the functional properties of genes and gene products across species. Each GO term is annotated with information, which includes the type of gene product (e.g., protein, tRNA, etc.) and an evidence code describing the type of evidence (e.g., experimental, phylogenetic, text mining, etc.). Annotations that are not curated manually are described as ‘IEA’ (inferred from electronic annotation). GO is structured as a graph comprising nodes representing each GO term. Edges between the nodes represent relationships between terms. The GO graph is hierarchical with ‘parent’ and ‘child’ terms, but unlike strict hierarchy, a given term may have more than one ‘parent’. Genes are annotated by a (a) molecular function, (b) cellular component, and (c) biological process. One of the most common methods to perform a GO term gene-set enrichment analysis is through the identification of common functions in a group of genes by means of an over-representation analysis (ORA). The method compares the GO terms corresponding to a test list of genes to the functions of selected background genes (for example, a database with functionally annotated genes). If the number of functions in the test list is greater than the number of functions obtained by chance, then these functions are considered to be over-represented.

A detailed description of the process including methodological challenges and pitfalls is given in relevant papers [65,66,67]. Several bioinformatics tools allow a gene enrichment analysis. Each has its own advantages but results can be unreliable depending on the coverage and resolution of the annotation databases they rely on [68,69]. The “stable” and “labile” CYPs were subjected to a GO term gene enrichment analysis using (a) STRING [70], in combination with g:Profiler [53], and (b) DAVID [54]. STRING/g:Profiler produced results for *D. melanogaster* “stable” and “labile” CYPs but failed to produce similar results for mosquito CYPs. Results obtained from DAVID showed enrichment in two GO terms, namely, the GO:0006082 term (organic acid metabolic process), and GO:0006700 (C21-steroid hormone biosynthetic process). The term GO:0006082 was enriched in 25 *A. aegypti*, *D. melanogaster*, *A. gambiae*, and *C. quinquefasciatus* “stable” genes. None of the “labile” CYPs showed enrichment in this term. GO:0006700 was found to be enriched for 10 “labile” proteins (5 each of *A. aegypti* and *D. melanogaster*) and 17 group 2 proteins belonging to *A. aegypti* (4), *D. melanogaster* (7), *A. gambiae* (3), and *C. quinquefasciatus* (4). The result indicates that the “stable” and “labile” list of P450s contain genes participating in the same biological processes.

To address the shortcomings of GO functional annotations, a number of predictive bioinformatic tools have been developed [56]. One of these, NetGO 3.0 [49], is considered to be one of the best at predicting function ontology [55]. The programme was used to predict enrichment of GO terms of the “stable” and “viable” P450s (Appendix A). “Stable” and “labile” proteins were differentially enriched in the terms shown in Table 9. Although individual proteins in both datasets can show enrichment in the same GO term, the relative proportions of proteins in each set are substantially different. Comparing the numbers of “labile” and “stable” P450, considerably more “stable” P450s showed enrichment of GO terms related to biosynthetic and developmental processes. This trend is reversed in the case of detoxification catabolic processes. GOATOOLS [57] was used to generate graphical views of enriched GO terms in biosynthetic, developmental, and catabolic processes. These are shown in Figure 3 and Figure 4, respectively.

## 4. Discussion

The non-culled dataset was split into two subgroups, which contained a total of 245 defined as phylogenetically “stable” and 241 as phylogenetically “labile”. The genes were obtained from the VectorBase database and cross-referred to Ensembl (Metazoa Genes 60 database). To avoid bias in statistical analyses, redundant proteins were removed from the datasets to generate non-redundant or culled datasets of “labile” and “stable” P450s. The culled and non-culled datasets were compared for a number of genomic and protein sequence properties. In general, sequence redundancy did not affect the overall picture and all sets follow similar statistical trends. Table 10 shows the gene and protein features that are strongly associated with “stable” P450 genes.

An analysis of genomic features of CYPs showed that “stable” genes are longer, have more exons, and have longer intron lengths. This is consistent with recent findings showing that longer genes correlate with functions that are important in the development stages of an organism [71,72]. Additionally, “stable” CYPs’ average intron length was shown to be almost twice the size of introns in “labile” P450s. This is in agreement with research showing that an intonic “burden” plays an important role in the evolutionary conservation of genes [73]. “Stable” CYPs were shown to have lower GC content than the “labile” ones and although the difference was not statistically significant, it is noted that GC content varies inversely with intron and exon length [74].

An analysis of physicochemical and protein sequence features revealed several differences between “labile” and “stable” P450s. “Labile” proteins were found to have a greater proportion of polar and charged residues and be less hydrophobic than “stable” P450s, which were relatively enriched in aliphatic amino acids. It is noted, however, that the method used to determine the relative hydrophobicity of P450s is a poor predictor as it does not take into account protein folding. It is known that the aggregation propensity of proteins is frequently linked to imperfect folding. In terms of amino acid composition, “stable” P450s were found to have Arg, Cys, Leu, and Trp in greater proportions than “labile” P450s. Arg is a known chaotrope associated with the prevention of protein aggregates and increasing the solubility of hydrophobes [75]. Recent studies have shown that Trp/Tyr chains are common in enzymes utilising O_2_ as a substrate and that redox active Trp/Tyr chains extend the functional lifetimes of P450s [76]. The relative enrichment of “stable” P450s in Cys residues may have a two-fold importance. On one hand, Cys residues may impose structural constraints necessary for specific, rather than promiscuous, protein–ligand interactions as in the case of interactions that are developmentally important [77]. On the other, Cys has a tendency to behave as a hydrophobic residue in folded proteins, despite possessing a polar sulfhydryl group [78].

Differences were observed in the amino acid composition of the haem-binding signature motif in the two datasets. In “labile” P450s, this motif is invariable, showing a total conservation of residues (PFSAGPRNCIGQRFA). In contrast, “stable” P450s show considerable variation in amino acid composition. By implication, the substrate specificity of the “stable” P450s would be expected to cover a much wider range of ligand structures.

A functional analysis of “stable” and “labile” P450s was studied by an enrichment analysis of GO terms. “Stable” P450s were predicted to be enriched in GO terms relevant to the biosynthesis of lipids and hormones. These biosynthetic processes are also essential for instar larval or pupal morphogenesis and play essential roles in the growth, reproduction, and defence systems of insects [79,80]. “Labile” P450s were predicted to be enriched in GO terms related to biological processes involving the detoxification of xenobiotics. As a cautionary note, it is pointed out that a GO gene enrichment analysis is not without methodological challenges and pitfalls [65,66,67] and results can be unreliable due to an inadequate coverage and resolution of the annotation databases [68,69].

The analysis of biological features, such as signal peptides and cellular localisation, was inconclusive and merits additional investigation. For example, the proportion of “labile” CYPs predicted to be located in the mitochondrion is higher than that for “stable” CYP, despite the fact that the absolute numbers are small to draw on definitive conclusions. This is contrary to phylogenetic studies showing that mitochondrial CYPs are relatively well conserved [25]. However, the name “mitochondrial” referring to the CYP clan should not be taken as definitive evidence of subcellular localisation and CYPs of this clan can be microsomal or have dual localisations [14]. Another important feature is the composition of transmembrane domains. The reducing power of CYPs depends on the coupling of the endoplasmic reticulum N-terminal transmembrane helix with membrane-anchored NADPH-dependent cytochrome P450 oxidoreductase [81]. A bioinformatic analysis of the sequence conservation of transmembrane anchors and the characterisation of motif patterns of signal and transit peptides can provide useful insights on the localisation and function of CYPs [82].

## 5. Conclusions

“Stable” and “labile” P450s were shown to be significantly different in a number of genomic and protein features. The interdependency of these features implies that multiple aspects of biology unite to determine whether a gene is “stable” or “labile” in mosquitos. The range of features analysed and discussed in this paper can be broadened and expanded to other organisms, thus gaining further insights into gene “essentiality”. Broadening the scope of the bioinformatic analysis presented in this paper may enable the classification of insect CYPs by training machine learning classifiers for the functional identification of insect P450s. Last but not least, the identification of “stable” genes associated with “lethality” may facilitate the identification of non-viable null mutants and thus the identification of new insecticide targets.

## Figures and Tables

**Figure 1 insects-16-00184-f001:**
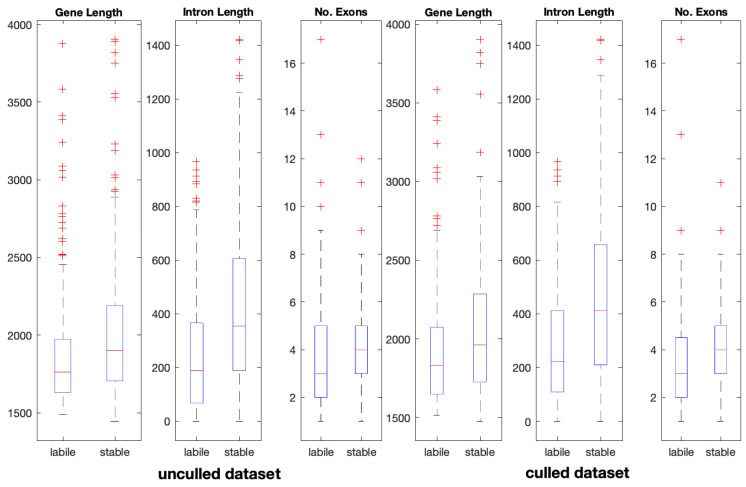
Distributions of the total gene length, number of exons, total length of exons, total length of introns, number of transcripts, and percentage of GC content in “stable” and “labile” genes. Outliers are depicted as (+). Median values are depicted by the red line in the boxplot.

**Figure 2 insects-16-00184-f002:**
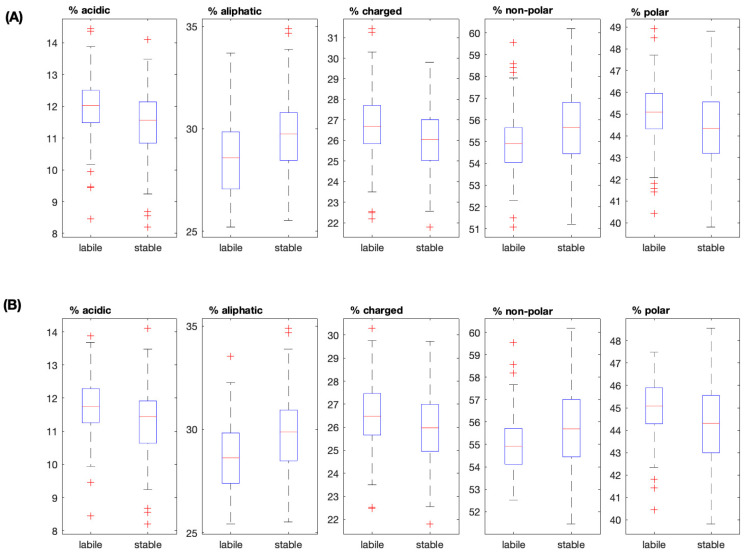
Distributions of acidic, aliphatic, charged, non-polar, and polar residues (%) between “labile” and “stable” P450s. Unculled and culled datasets denoted as (**A**,**B**), respectively. Outliers are depicted as (+). Median values are depicted by the red line in the boxplot.

**Figure 3 insects-16-00184-f003:**
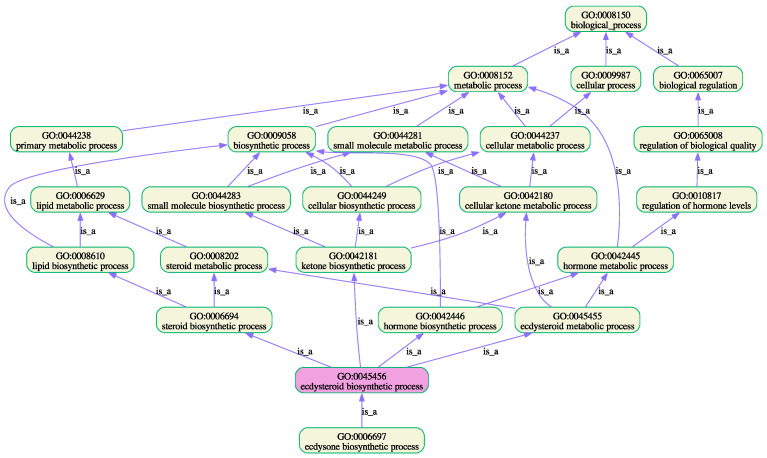
A plot of the enriched GO terms and their ancestors (Table 7, GO terms 1–8).

**Figure 4 insects-16-00184-f004:**
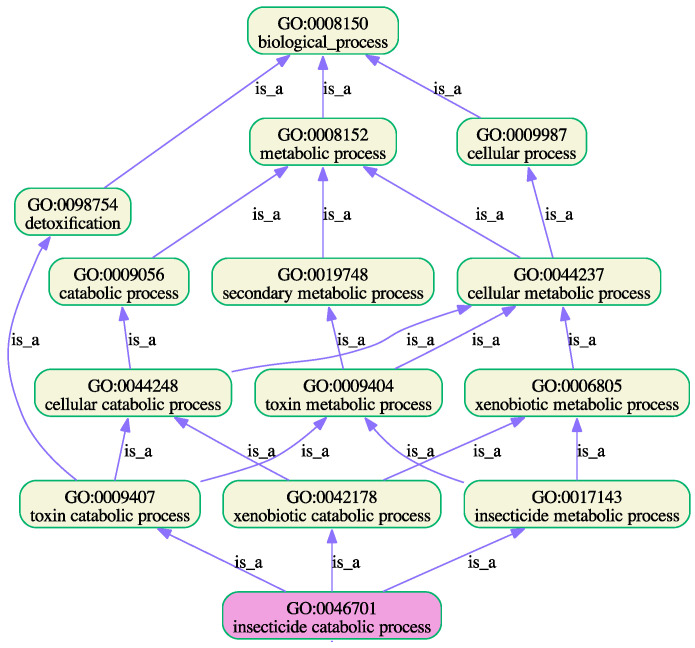
A plot of the enriched GO terms and their ancestors (Table 7, GO terms 9–11).

**Table 1 insects-16-00184-t001:** P450 gene numbers and their clan-wise distribution in *D. melanogaster* (*Dmel*), *A. gambiae* (*Agam*), *A. aegypti* (*Aaeg*), and *C. quinquefasciatus* (*Cqui*).

	Total	Clan 2	Clan 3	Clan 4	Mitochondrial	Source
		Family	Family	Family	Family	
*Dmel*		CYP18, 303–307	CYP6, 9, 28, 308–310, 317	CYP4, 311–313, 316, 318	CYP12, 49, 301–2, 314–5	
	85	No. of genes: 6	No. of genes: 36	No. of genes: 32	No. of genes: 11	[14]
*Agam*		CYP15, 303–307	CYP6, 9, 329	CYP4, 325	CYP12, 49, 301–2, 314–5	
	106	No. of genes: 10	No. of genes: 42	No. of genes: 45	No. of genes: 9	[14]
*Aaeg*		CYP15, 18, 303–307	CYP6, 9, 329	CYP4, 325	CYP12, 49, 301–2, 314–5	[21]
	164	No. of genes: 11	No. of genes: 84	No. of genes: 59	No. of genes: 10	
*Cqui*		Cyp15, 303–307	CYP6, 9, 329	CYP4, 325	Cyp12, 301–2, 314–14	
	196	No. of genes: 14	No. of genes: 88	No. of genes: 83	No. of genes: 11	[22]

**Table 2 insects-16-00184-t002:** Number of CYP genes and proteins in “labile” and “stable” datasets.

		Unculled		Culled	
	No. of Genes	“Stable”	“Labile”	“Stable”	“Labile”
*Dmel*	83	53	30	46	22
*Agam*	94	49	45	31	28
*Aaeg*	131	58	73	33	30
*Cqui*	178	85	93	52	35
Total	486	245	241	162	115

**Table 3 insects-16-00184-t003:** Sequence and functional features and corresponding bioinformatic tools.

Features	Bioinformatic Methods
Genomic features: gene length, % of GC content, number of transcripts, number of exons, length of exon and intron	VectorBase [41]
Protein sequence features: protein length, molecular weight, protein charge, isoelectric point, amino acid composition, hydrophobicity	EMBOSS Pepstats [46]
Phosphorylation	PhosNet 3.0 [47]
Signal peptide; Transmembrane domains; Subcellular localisation	BUSCA [48]
Gene Ontology terms: biological process, cellular component, molecular function	NetGO 3.0 [49]

**Table 4 insects-16-00184-t004:** Median gene length, GC contents, number of transcripts, number of exons, exon length, and intron length for essential and viable genes *.

Datasets		Gene Length (bp)	No. of Exons	Exon Length (bp)	Intron Length (bp)	No. of Transcripts	% GC Content
**Non-culled**	“labile”	1840	3	1524	311	1	47.20
	“stable”	2149	4	1518	612	1	44.78
	*p*-value	**2.8034 × 10^−7^**	**1.0212 × 10^−8^**	0.0209	**7.6899 × 10^−9^**	0.3531	0.0270
**Culled**	“labile”	1890	3	1527	359	1	46.48
	“stable”	2165	4	1521	612	1	45.63
	*p*-value	**7.1450 × 10^−4^**	**4.4157 × 10^−6^**	0.0302	**1.5198 × 10^−4^**	0.6973	0.3718

* The median value of each feature is reported. *p*-Values were determined from a Mann–Whitney U test. Statistically significant results were evaluated based on the Bonferroni corrected *p*-value of 0.0083. They are shown in bold typeface.

**Table 5 insects-16-00184-t005:** Median values of different protein features and the *p*-values of their distribution calculated using the Mann–Whitney U test *.

	Unculled			Culled		
Property	“Labile”	“Stable”	*p*-Value	“Labile”	“Stable”	*p*-Value
Molecular weight	5.8236 × 10^4^	5.8106 × 10^4^	0.0124	5.8454 × 10^4^	5.8162 × 10^4^	0.0249
Isoelectric point	8.3088	8.1503	0.2324	8.2718	8.3542	0.4991
Charge	9	9.5000	0.4037	10	10.5000	0.0724
Hydrophobicity	−19.6450	−16.0583	**4.3384 × 10^−8^**	−18.7226	−15.5340	**2.7720 × 10^−4^**
Aromatic	13.2110	13.1148	0.5403	13.2110	13.1417	0.6721
Aliphatic	28.5714	29.7619	**3.4826 × 10^−10^**	28.6299	29.8651	**3.1629 × 10^−6^**
Acidic	12.0240	11.5686	**3.5141 × 10^−8^**	11.7530	11.4458	**3.8132 × 10^−5^**
Basic	14.7810	14.6000	0.0258	14.6535	14.6939	0.6806
Charged	26.6791	26.0521	**1.2929 × 10^−6^**	26.4706	25.9669	0.0056
Polar	45.0980	44.3340	**6.3566 × 10^−7^**	45.0902	44.3137	**2.4062 × 10^−4^**
Non-polar	54.9020	55.6660	**5.1469 × 10^−7^**	54.9098	55.6863	**2.4062 × 10^−4^**
Small	44.6000	44.6939	0.6356	44.6680	44.4890	0.6232
Tiny	23.3202	23.5887	0.0306	23.5409	23.8095	0.3240

* The median value of each feature is reported. *p*-Values were determined from a Mann–Whitney U test. Statistically significant results were evaluated based on the Bonferroni corrected *p*-value of 0.0038. They are shown in bold typeface.

**Table 6 insects-16-00184-t006:** Differences in the amino acid frequency of usage between the P450 proteins of the two groups in the culled and non-culled datasets.

	Unculled			Culled		
aa *	“Labile”	“Stable”	*p*-Value	“Labile”	“Stable”	*p*-Value
Ala	5.6711	6.1100	**4.0438 × 10^−4^**	5.8601	6.2500	0.0145
Cys	1.1811	1.5238	**1.9760 × 10^−10^**	1.2048	1.5385	**2.8718 × 10^−5^**
Asp	5.4409	5.3465	0.0230	5.4104	5.2427	0.0963
Glu	6.4338	6.1151	**8.9284 × 10^−6^**	6.3241	6.0362	**0.0012**
Phe	6.5476	6.2000	**0.0014**	6.4833	6.1100	0.0150
Gly	5.6075	5.3254	0.0125	5.6863	5.3407	0.0966
His	2.1696	2.3301	**4.2799 × 10^−5^**	2.2018	2.3297	0.0233
Ile	6.1185	6.0827	0.8018	6.1185	6.0038	0.7398
Lys	6.4639	5.4000	**1.3464 × 10^−14^**	6.0998	5.3360	**3.4017 × 10^−5^**
Leu	10.0616	11.0656	**2.6111 × 10^−12^**	10.2970	11.0891	**7.5251 × 10^−7^**
Met	3.3730	2.9851	**4.2922 × 10^−5^**	3.4068	3.0364	**0.0015**
Asn	4.0161	3.9448	0.3603	3.9062	3.8076	0.1330
Pro	5.0710	5.1081	0.2145	4.9900	5.0813	0.1682
Gln	3.4765	3.5849	0.1371	3.6290	3.6735	0.5681
Arg	6.0852	6.6202	**1.0888 × 10^−4^**	6.3116	6.7308	**0.0013**
Ser	5.3435	5.4104	0.4267	5.3465	5.4409	0.4281
Thr	5.4326	5.2104	**0.0015**	5.4902	5.1383	**0.0011**
Val	6.5056	6.2745	0.0128	6.4885	6.2622	0.0318
Trp	0.9452	1.1236	**4.8193 × 10^−6^**	0.9328	1.1494	**3.9031 × 10^−5^**
Tyr	3.5185	3.6072	0.5588	3.4926	3.6000	0.5164

* Amino acid (aa). The *p*-value for the Bonferroni correction is 0.0025. Statistically significant differences are shown in bold typeface.

**Table 7 insects-16-00184-t007:** Summary of predicted signal peptides, subcellular localisation, and transmembrane helices.

	Non-Culled		Culled	
Biological Feature	No. of “Labile” CYPs	No. of “Stable” CYPs	No. of “Labile” CYPs	No. of “Stable” CYPs
Signal peptide	19 (7.9%)	28 (11.4%)	8 (7.0%)	18 (11.2%)
Mito transit	11 (4.6%)	5 (2%)	7 (6.1%)	2 (1.3%)
Mitochondrial membrane	11 (4.6%)	5 (2%)	7 (6.1%)	2 (1.3%)
N-terminal helix	198 (82.2%)	196 (80.0%)	95 (82.6%)	132 (81.5%)

**Table 8 insects-16-00184-t008:** The conservation of the PERF and haem-binding signature motifs in “labile” and “stable” P450s.

Motif	“Stable” (Unculled)	“Labile” (Unculled)
Haem-binding	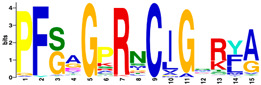	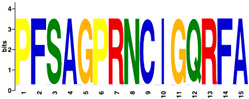
PERF	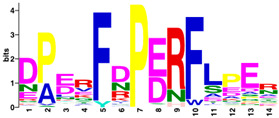	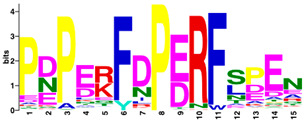
**Culled**	**“Stable” (Culled)**	**“Labile” (Culled)**
Haem-binding	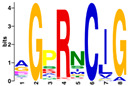	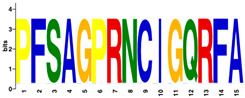
PERF	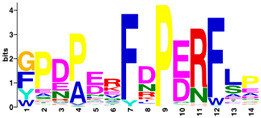	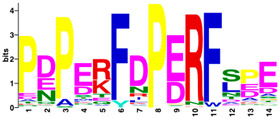

**Table 9 insects-16-00184-t009:** Gene enrichment in GO terms related to biosynthetic, developmental, and detoxification metabolic processes.

GO Term	Description	Subset ^1^	All CYPs ^2^	*Dmel* ^2^	*Agam* ^2^	*Aaeg* ^2^	*Cqui* ^2^
1. GO:0048856	anatomical structure development	S	135 (55.1)	27 (11.0)	26 (10.6)	33 (13.5)	49 (20)
		L	75	7	18	23	27
2. GO:0008610	lipid biosynthetic process	S	141 (57.5)	26 (10.6)	26 (10.6)	38 (15.5)	51 (20.8)
		L	24 (9.9)	4 (1.7)	3 (1.2)	8 (3.3)	9 (3.7)
3. GO:0008202	steroid metabolic process	S	95 (38.8)	11 (4.5)	29 (11.8)	37 (15.1)	18 (7.3)
		L	10 (4.1)	2 (0.8)	1 (0.4)	1 (0.4)	6 (2.5)
4. GO:0042445	hormone metabolic process	S	53 (21.6)	12 (4.9)	15 (6.1)	15 (6.1)	11(4.5)
		L	4 (1.6)	-	-	-	4 (1.6)
5. GO:0007275	multicellular organism development	S	91 (37.1)	19 (7.7)	18 (7.3)	22 (9.0)	32 (13.0)
		L	25 (10.4)	2 (0.8)	7 (2.9)	9 (3.7)	7 (2.9)
6. GO:0009791	post-embryonic development	S	19 (7.7)	6 (2.4)	4 (1.6)	6 (2.4)	3 (1.2)
		L	-	-	-	-	-
7. GO:0002165	instar larval or pupal development	S	17 (6.9)	6 (2.4)	6 (2.4)	3 (1.2)	2 (0.8)
		L	-	-	-	-	-
8. GO:0045456	ecdysteroid biosynthetic process	S	10 (4.1)	4 (1.6)	4 (1.6)	1 (0.4)	1 (0.4)
		L	-	-	-	-	-
9. GO:0006805	xenobiotic metabolic process	S	24 (9.8)	2 (0.8)	5 (2.0)	6 (2.4)	11 (4.5)
		L	69 (28.6)	9 (3.7)	11 (4.6)	28 (11.6)	21 (8.7)
10. GO:0046680	response to DDT	S	30 (12.2)	8 (3.3)	5 (2.0)	8 (3.3)	9 (3.7)
		L	86 (35.7)	10 (4.1)	22 (9.1)	29 (12.0)	25 (10.4)
11. GO:0009404	toxin metabolic process	S	12 (4.9)	3 (1.2)	2 (0.8)	2 (0.8)	5 (2.0)
		L	28 (11.6)	7 (2.9)	2 (0.8)	10 (4.1)	9 (3.7)

^1^ S denotes “stable”; L denotes “labile”. ^2^ Values in () denote %.

**Table 10 insects-16-00184-t010:** Summary of characteristics likely to be associated with essential or viable genes.

“Stable” Gene Tendencies	“Labile” Gene Tendencies
Longer genes; longer introns; more exons	Simpler, shorter gene structure
More hydrophobic; relative proportion of aliphatic amino acids higher; enriched in Cys, Arg, Leu, and Trp	Less hydrophobic; relative proportion of charged and polar amino acids higher; enriched in Glu, Lys, and Met
Involved in biosynthetic and developmental processes, such as biosynthesis of lipids and hormones essential for instar larval or pupal morphogenesis	Involved in cellular catabolic processes, detoxification of xenobiotics, and insecticide metabolic processes

## Data Availability

Additional data available upon request.

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
