# Peer review of "Properties of “Stable” Mosquito Cytochrome P450 Enzymes"

_insects, 2025, doi:10.3390/insects16020184_

Round 1
Reviewer 1 Report
Comments and Suggestions for Authors
The manuscript titled "Properties of “stable” mosquito cytochrome P450 enzymes" investigates whether there are differences between stable and labile P450 genes at the sequence level across several mosquito species. To this end, P450 genes sequences of three mosquito species and D. melanogaster were retrieved from public databases and compared for the following criteria:
protein sequence properties (molecular weight, number of residues, charge, isoelectric point, amino acid composition, hydrophobicity)
genomic properties (gene length, cds length, intron/exon length, %GC content)
phosphorylation
signal peptides, transmembrane domains and protein localisation
gene ontology terms
While the manuscript is overall well written and structured, I have several comments for consideration:
Major comments:
While a comparison between four dipteran species is good, a broader comparison across different insect orders would be even better given that the occurence of labile and stable P450 lineages is not restricted to dipterans. At least, a comparison with a non-dipteran outgroup would be beneficial to see if observed differences are broadly applicable or only within this order.
Several of the assessment criteria are dependent on the quality of the genomic sequences used. Can you comment in which way a manual curation of e.g. intron/exon boundaries, gene length etc. has been undertaken either by yourself or the databases where the sequences were retrieved from.
The significant differences of protein features between the two groups is interesting. I wonder whether a comparison across the whole protein sequence is sufficient, though. Have you considered looking at differences of protein features or amino acid residue frequences at specific parts of the enzymes (e.g. residues lining the catalytic pocket or residues associated with the substrate recognition sites of P450s)?
It appears odd that two transmembrane regions were predicted for all P450s and 80% of the P450s are supposedly lacking the N-terminal helix which is in contrast to most of the available literature. How can this be explained?
I am skeptical about the GO annotation analysis, but this may be my lack of understanding of the methodology using protein language models. Can you provide more background what proportion of annotations used in your analysis are based on experimental evidence or just sequence similarity/phylogenetic relation?
Minor comments:
Introduction:
L77: phase III instead of IIII.
LL119-122: these blooms have originated long before the first exposure to chemical insecticides applied in modern agriculture. Please revise this sentence.
Methods:
L174: Is there any reasoning for the threshold of 70%?
Results:
Throughout all tables and figures, there is always both datasets present - culled and unculled. I barely see a difference between those two and wonder if it is more concise to focus on the culled dataset only given that you highlight in the methods section that the non-redudant dataset is necessary to "avoid undesirable bias in statistical analysis".
Author Response
Thank you for your useful comments. My response to each comment is included in the attached file.

Reviewer 2 Report
Comments and Suggestions for Authors
The manuscript highlights the role of enzymes involved in detoxification processes, specifically P450, which are involved in the degradative metabolism of various insecticidal groups. In this work, the classification made by the authors is very useful to mediate strategies for both vector control and resistance management. I recommend that this be accepted with some specific changes.
Simple summary. - Must be reduced, it is larger than the abstract.
Keywords: Change P450
The first paragraph doesn’t have citations.
Table 5. specified what is MW and IEP
Line 265, place the full name of the amino acids (abbreviation) and in subsequent mentions its abbreviation.
Table 6.- Change Aa for aa
Table 8.- in the footer change labile) for labile“
Change the punctuation 10. GO:0046680 response to DDT S 30 (12.2) for 12.2
Author Response
Thank you for your comments. My responses follow below:
Comment1. Simple summary. - Must be reduced, it is larger than the abstract.
Response. ReducedWord count (simple summary): 165 words. Work count (abstract): 183
Comment 2. Keywords: Change P450.
Response: Changed to cytochrome P450 enzymes
Comment 3. The first paragraph doesn’t have citations.
Response: Three citations introduced
Comment 4. Table 5. specified what is MW and IEP.
Response: Changed to “molecular weight” and “isoelectric point”.
Comment 5. Line 265, place the full name of the amino acids (abbreviation) and in subsequent mentions its abbreviation.
Response: Changed accordingly
Comment 6. Table 6.- Change Aa for aa.
Response: Changed
Comment 7. Table 8.- in the footer change labile) for labile“.
Response: Changed
Comment 8. Change the punctuation 10. GO:0046680 response to DDT S 30 (12.2) for 12.2.
Response: Changed
Reviewer 3 Report
Comments and Suggestions for Authors
This manuscript (insects-3395615) explored the classification of cytochrome P450 enzymes (CYPs) in mosquitoes, the differences between “stable” and “labile” CYPs based on genomic and sequence features, and the implications of these differences for mosquito physiology and detoxification of xenobiotics as well as insecticide metabolic. The manuscript is written in a scientific manner, and the data sufficiently supports its main conclusion. In its current form, it meets the criteria for publication in the journal. However, some questions should be addressed before publication.
1. There are many verified P450 genes related to resistance and many verified genes related to growth and development. And analyze these genes to see if their characteristics are consistent with the model? It would be more convincing.
Author Response
Thank you for your comments. My response follows below:
Comment 1. There are many verified P450 genes related to resistance and many verified genes related to growth and development. And analyze these genes to see if their characteristics are consistent with the model? It would be more convincing.
Response: The point is well taken. However, in my experience, with the exception of the fruit fly, they are not “many verified genes related to growth and development”. Furthermore, there are too many unanswered questions. Quoting from Vontas et al. “Fundamental questions, such as: i) the actual contribution of an individual CYP alone or in combination with other detoxification pathway components, or other mechanisms (target site, absorption), in the resistance phenotype; ii) the regulation of the elevated CYPs in resistance mosquitoes; iii) the molecular physiology and localization of CYP detoxification enzymes; iv) the association between overexpression levels and allelic variation with catalytic activity and the intensity of insecticide resistance phenotype (resistance ratio), as well as v) the true value of molecular diagnostics targeting CYP markers for Insecticide Resistance Management (IRM), have not been answered. Vontas et al. Cytochrome P450-based metabolic insecticide resistance in Anopheles and Aedes mosquito vectors: Muddying the waters. Pestic Biochem Physiol, 2020. 170: p. 104666.
Last but not least, I would like to point out that the major objective of analysis reported in this paper is to find out if one could identify genomic and chemical features distinguishing “stable” from “labile” CYPs. The analysis is intended to serve as a proof-of-concept for developing a machine learning classifier for “essential”/“non-essential” genes.
Round 2
Reviewer 1 Report
Comments and Suggestions for Authors
The comments have been adequately adressed by the author and the study is worthy of publication.